# OpenReview forum: "Path-Consistency with Prefix Enhancement for Efficient Inference in LLM"
_ICLR.cc/2026/Conference — Submitted to ICLR 2026_

### Official Review · Reviewer_XE47 · 2025-10-29

**Soundness:** 2
**Presentation:** 2
**Contribution:** 2
**Rating:** 2
**Confidence:** 4

**Summary:**

This paper proposes path-consistency, an improved variant of self-consistency that aggregates reliable prefixes across a small batch of completions to reduce the token cost during sampling. The authors show that this method can improve performance and reduce inference latency on several datasets using Llama3-8B.

**Strengths:**

**[S1]** The idea is reasonable. By extracting reliable prefixes, the proposed method reduces token consumption during self-consistency sampling, thus lowering inference time.

**[S2]** According to the provided experiments, the method improves performance while decreasing latency.

**Weaknesses:**

**[W1]** The paper lacks comparison with recent works; most of the cited references are outdated and do not include any from 2025.

**[W2]** The main experiments are conducted only on Llama3-8B, which raises concerns about robustness. The authors should extend their evaluation to other model families and sizes.

**[W3]** The derivation in Section 4.2 relies on too many assumptions, making the theoretical justification unconvincing.

**[W4]** The reported self-consistency accuracy on GSM8K (64.1% in Table 2) is significantly lower than the official Llama3 result (~80%). This discrepancy questions the soundness of the experimental setup.

**[W5]** The paper does not evaluate on more challenging reasoning datasets such as MATH500 or AIME, which are commonly used and better reflect the method’s robustness on difficult multi-step reasoning tasks.

**[W6]** The claim around lines 324–325 lacks empirical support. The authors should include an ablation study on the sampling path parameter.

**[W7]** As shown in Tables 5 and 6, the proposed method shows limited advantage compared to related methods.

**[W8, minor]** The captions of Figures 1 and 2 exceed the bottom page margin.

**Questions:**

**[Q1]** Could the authors clarify how inference latency is measured? This is critical for assessing the efficiency of the proposed method.

**[Q2]** The authors do not report latency metrics for evaluations using DeepSeek-V3 with Llama-3.2-1B-Instruct. Could they provide corresponding results?

---

> ### Author Response · Authors · 2025-11-21
> **Response to Reviewer XE47**
>
> Dear Reviewer XE47,
>
> Thank you for the detailed feedback! We appreciate the opportunity to address your concerns and improve our work.
>
> > **Q1** & **Q2**: Inference latency measurement
>
> All inference latency is measured **locally** using the Meta Llama 3 official Python API (`Llama.build(...)`).  For each branch, we record the end-to-end generation time returned by the model in its metadata:
> ```python
> results = generator.text_completion(...)
> o_times.append(results[-1]["time"])
> ```
> For completeness, here we report absolute inference time (in seconds) for local models Llama3-8B and Llama-3.2-1B-Instruct on GSM8K. All measurements are obtained using Meta’s official Llama runtime on a single NVIDIA A40 GPU:
> | Model|SC|PC|AC|ESC|
> |-|-|-|-|-|
> |Llama3-8B|3.45|2.95|3.72|3.71|
> |Llama-3.2-1B-Instruct|1.09|0.91|1.08|1.09|
>
> Efficiency  is primarily measured using token consumption, a stable and model-independent metric that accurately reflects inference efficiency while avoiding hardware-related fluctuations.  Local latency measurements can be heavily affected by many uncontrolled factors, including hardware specifications, current system workload, memory/IO contention, and data transfer overhead. For API-based models such as DeepSeek, latency becomes even less reliable due to additional variability from network conditions and service-side scheduling.
>
> For this reason, although we record latency during experimentation, our paper evaluates efficiency primarily through reduction in generated token count. Token-level savings provide a hardware-agnostic and implementation-independent measure of acceleration. Therefore, reporting token reduction allows us to fairly and consistently quantify the efficiency gains of Path-Consistency, independent of the execution environment.
>
> > **W2**: Limited evaluation scope
>
> To demonstrate robustness and model-agnostic behavior, we evaluated PC on multiple models: Llama3-8B, Llama-3.2-1B-Instruct (Table 7), and DeepSeek-V3 (Table 6). Across these diverse model families and sizes, PC consistently maintains efficiency gains without degrading task accuracy. In addition, we also conducted related experiments on other model families such as mistral-7B:
>
> |Method|C_threshold|Accuracy(%)|Speedup(%)|Decrease(%)|
> |-|-|-|-|-|
> |SC|-|59.7|-|-|
> |**PC**|0|58.5|+22.0|-20.3|
> ||0.8|60.3|+11.1|-11.4|
> ||0.9|60.4|+6.2| -6.6         |
>
> > **W3**: Unconvincing theoretical justification
>
> The assumptions in Section 4.2 are as follows:
>
> - The model generates at most one dominant wrong answer.
> - The model performs better than random.
>
> These are mild and meaningful assumptions: the first represents the provable worst-case scenario, as distributing probability mass among multiple wrong answers only lowers the risk of mis-promotion; the second is reasonable since PC focuses on improving reasoning efficiency given a functional model. We provide a formal proof in the appendix to clarify this point.
>
> > **W4**: Experimental discrepancy
>
> As described in Sec. 5.1, our experiments use the open-source base model Llama3-8B. In contrast, the official GSM8K result (\~80%) reports the performance of the instruct-tuned Llama3-8B. Therefore, the observed accuracy difference (\~64.1%) in our setup) is expected and reflects the distinction between base and instruct-tuned models. Moreover, our results are obtained using the **official Meta released prompt** and evaluated via **local inference** on the publicly available Llama3-8B base model. Thus the reported accuracy is the *actual, reproducible performance* of the base model.
> > **W5**: Lack of evaluation on challenging datasets
>
> We have added experiments on AIME2025 with DeepSeek-V3 to assess PC on more challenging datasets:
> | Method | Accuracy (%) | Decrease (%) |
> |-|-|-|
> |SC|60.0|-|
> |PC|76.7|-4.1%|
>
>  The results show that while PC’s speedup effect remains present on long chains, it is less pronounced, achieving a 4% reduction in token consumption. Meanwhile, there is a substantial improvement in accuracy when applying PC, because of PC filtering out early unstable reasoning branches and enforcing prefix agreement, which helps long CoT sequences converge to more coherent and correct solutions. Thank you for the helpful suggestion.
> > **W6**: Unsupported claim
>
> The effect of branch count on SC performance is well-studied (Fig. 2 in Wang et al., 2023), showing saturation at ~20 branches. We follow that established operating point.
> > **W7**: Limited advantage over related methods
>
> PC primarily targets efficiency in chain-of-thought reasoning. On Llama3-8B, PC reduces token consumption by ≥57.8% with at most 1.2% accuracy drop. On DeepSeek-V3, PC reduces token usage by 5.34–41.7% while maintaining accuracy, showing consistent gains in practical reasoning scenarios.
>
> **References**:
> Wang, X., Wei, J., Schuurmans, D., Le, Q., Chi, E., Narang, S., Chowdhery, A., & Zhou, D. (2023). Self-consistency improves chain of thought reasoning in language models. arXiv preprint arXiv:2203.11171.

---

> ### Comment · Reviewer_XE47 · 2025-11-26
>
> I would like to thank the authors for their reply. But I am still confused about some points.
>
> > Q1 & Q2: our paper evaluates efficiency primarily through reduction in generated token count.
>
> Personally, I think latency serves as a better metric for efficiency estimation in this paper. SC/PC is not a single-thread inference procedure, which means for a user query, the method may generate more than one completion, or we need to pre-process or post-process based on the completions group. So maybe at an inference step, the fast-completed completions need to wait for the slow-completed completions, but metric of token number counts remove this effect.
>
> For end users, we exactly do not care about the token counts, instead, we need both accurate and fast-to-respond features.
>
> > W6: The effect of branch count on SC performance is well-studied
>
> From my perspective, LLMs have made a big difference from 2023, we can see that in many tech reports, they will release cons@64 or avg@64 metrics, showing that 20 is not a converged number for SC or maybe PC.
>
> ---
>
> Besides, please reflect your new analyses, experimental results and formatting corrects in the rebuttal revision, as I cannot see them in the current pdf.

---

> > ### Author Response · Authors · 2025-11-28
> >
> > Thank you for your time and constructive feedback. I will address your concern points in detail below.
> >
> > > Latency vs Token consumption
> >
> > We do not claim that latency is unimportant. This is why we also report latency in our experiments in Table 2, but our analysis primarily focuses on token consumption, which has an almost linear relationship with latency while avoiding the instability caused by system-level factors (hardware load, kernel scheduling variability, network jitter, etc.). As such, **token consumption faithfully captures the core mechanism that determines latency**, in a reproducible and implementation-agnostic way.
> >
> > SC/PC is fundamentally a **decoding strategy** rather than a **fixed inference procedure**. It can be implemented either sequentially or in parallel depending on the underlying inference engine. Therefore, SC/PC itself is not inherently “single-threaded” or “multi-threaded”; any threading behavior arises purely from the implementation rather than from the algorithmic design.
> >
> > Similarly, in Aggarwal et al. (2023), Adaptive-Consistency is formulated as an **incremental sampling procedure**: after each newly generated sample, the algorithm updates the vote distribution and applies a stopping criterion to decide whether to request an additional sample or terminate. Conceptually, this makes the sampling process sequential over samples, even though the paper does not propose any specialized parallel decoding architecture. Any parallelism (e.g., batched API calls) is an implementation detail and is orthogonal to the AC algorithm itself.
> >
> > Our work follows the same principle. We focus on improving **algorithmic efficiency**, and users or deployment platforms remain free to implement the algorithm in either a “single-threaded” mode or a “multi-threaded” mode based on the chosen window size. This flexibility makes latency a poor indicator of the algorithm’s intrinsic efficiency, since latency is heavily influenced by implementation choices.
> >
> > In a “multi-threaded” setting, as you pointed out, fast-finishing completions may need to wait for slow-finishing ones. However, because parallelism is bounded by the window size, and all completions within a window share the same prefix depth, the differences in completion length are limited. Therefore, **token consumption remains a stable and meaningful metric**, and does not break under parallel execution. Moreover, reductions in generated tokens directly correspond to reductions in actual computational cost, energy usage, and even API billing cost.
> >
> > In summary, while latency can fluctuate depending on implementation and system conditions, token consumption provides a reliable, reproducible, and architecture-independent measure of the true algorithmic efficiency, which is why our analysis centers on this metric.
> >
> > > Branch Count
> >
> > We agree that 20 samples is not necessarily a universal convergence point for all models, datasets, or decoding strategies. That said, our experimental setting with 20-path SC is still a valid and meaningful evaluation. More importantly, PC is designed as an **efficiency** method. Using 20 samples as the baseline is actually a stricter and more challenging evaluation for PC:
> >
> > - The fewer branches SC uses, the harder it is for PC to match SC accuracy while reducing sampling.
> >
> > - Conversely, evaluating PC against SC with more samples (e.g., 32, 64) would make PC’s job easier, since SC becomes more redundant and PC can more effectively exploit prefix-level consistency.
> >
> > Therefore, while PC might show reduced benefits when SC uses very few samples, its effectiveness is guaranteed at larger sample sizes, and may even improve further. Our results at 20 samples thus demonstrate the robustness of PC under a conservative, challenging baseline rather than relying on an inflated number of branches.
> >
> > We have added experiments using 40-path SC and PC on GSM8K with Llama-8B:
> >
> > | Method | Accuracy (%) | Tokens (# per problem) |
> > | ------ | ------------ | ---------------------- |
> > | SC-40  | 68.9         | 3861                   |
> > | PC-40  | 68.5         | 1988 (-48.5%)          |
> >
> > The relative benefit persists even as the teacher improves or sample budget increases.
> >
> > ---
> >
> > The supplemental results and theoretical analysis are now included in the appendix.

---

> > > ### Comment · Reviewer_XE47 · 2025-11-28
> > >
> > > I appreciate the authors' new response. I am happy to see the new contents in the revision. I would like to increase my score to reflect my current evaluation of this paper.
> > >
> > > Please note that, the "EDIT" button is now not visible on the OpenReview platform, so I cannot modify the score at this time. Feel free to remind me if the "EDIT" button is available, and I will also keep an eye.

---

### Official Review · Reviewer_SuGi · 2025-10-31

**Soundness:** 3
**Presentation:** 3
**Contribution:** 3
**Rating:** 6
**Confidence:** 4

**Summary:**

This paper presents path-consistency, with the aim to reduce the token consumption or improve the efficiency of standard self-consistency. The key idea is to extract prefixes from early branches and use them to guide the subsequent generations. Experiments are conducted on three domains of datasets. The results show that the proposed method could significantly improve the efficiency while preserve the accuracy in most cases.

**Strengths:**

- The paper is generally well-written.
- The proposed path-consistency is reasonable.
- Extensive experiments are conducted to show the effectiveness of the proposed method.

**Weaknesses:**

- In the experiments, the authors show that there exists a proper confidence threshold at which path-consistency simultaneously enhances both task performance and efficiency. However, how to determine this threshold is unclear.

- The authors claim that the proposed method integrates seamlessly with existing optimization methods, achieving even better acceleration performance. However, only self-consitency is evaluated in the experiments.

- Related to the previous point, only self-consistency is evaluated as baselines in the experiments. How about the direct generation methods as well as inference time techniques. An accuracy-efficiency comparison among more related techniques is appreciated.

**Questions:**

- The proposed method largely relies on the correctness of the prefixs of earlier generated paths. Can the authors analyze the results to check how the correctness of the prefixs affect the results?

---

> ### Author Response · Authors · 2025-11-21
> **Response to Reviewer SuGi**
>
> Dear Reviewer SuGi,
>
> Thank you for the detailed and constructive feedback! We appreciate the opportunity to address your concerns and improve our work.
>
> > **Q1**: Effect of correctness of prefixes
>
> Iterative prefix selection **does not increase the risk of propagating early minority errors**. In fact, the probability that an incorrect prefix passes multiple Beta gates decreases exponentially with depth, since each gate independently requires a majority concentration above the threshold. We include a formal proof in the **Supplementary Theoretical Analysis** and will be added to the appendix to make this behavior explicit.
>
> Meanwhile, we analyzed the per-branch behavior of PC and found that prefix correctness does not need to be perfect in early iterations, as the method naturally performs self-correction when early prefixes become unsupportive. Let's see this case:
>
> ```json
> {"input":"Two girls each got 1\/6 of the 24 liters of water. Then a boy got 6 liters of water. How many liters of water were left?","target":10}
> {"id": 80, "result": 1, "target": 10, "answer": 10.0,
> "answers": [8.0, 10.0, 14.0, 14.0, 14.0, 2.0, 14.0, 14.0, 10.0, 16.0, 10.0, 10.0, 10.0, 10.0, 12.0, 16.0, 14.0, 2.0, 10.0, 14.0],
> "tokens_gen": [70, 90, 63, 76, 84, 52, 57, 59, 54, 61, 15, 15, 15, 15, 29, 68, 65, 76, 94, 77], "
> level": [0, 0, 0, 0, 0, 1, 1, 1, 1, 1, 2, 2, 2, 2, 2, 0, 0, 0, 0, 0],
> ```
>
> In this GSM8K instance using the Llama-3.2-1B-Instruct model, the first several sampled branches do not converge to the correct answer, and the initially extracted prefix is itself incorrect. However, as the sampling branches expands, the distribution shifts, the confidence of the current prefix falls below the Beta-gated threshold, and PC **automatically discards** the erroneous prefix. This “prefix correction” step allows later branches to align toward the correct reasoning (final answer: 10 liters), despite early inconsistencies.
>
> > **W1**: Determination of confidence threshold
>
> The confidence threshold control the efficiency–accuracy trade-off:
>
> - Smaller thresholds --> more aggressive extraction (higher speedup).
> - Larger thresholds --> more conservative extraction (higher accuracy).
>
> Notably, when the threshold is less than 0.5, PC reduces to majority voting, giving a clean conceptual baseline.
>
> > **W2&3**: Limited baselines
>
> We compared PC against multiple inference optimization techniques, including self-consistency (SC) Wang et al. (2023), Adaptive Consistency (AC) Aggarwal et al. (2023), and Early-Stopping Self-Consistency (ESC) Li et al. (2024). Across various datasets and models, PC consistently achieves task accuracy comparable to or slightly better than other methods while consuming significantly fewer tokens. We have included the results on datasets GSM8K, SVAMP, and StrategyQA with models Llama3-8B, DeepSeek-V3, and Llama-3.2-1B-Instruct, in Table 5, 6, and 7, respectively.
>
> Specifically, the data shows:
>
> - **Competitive Accuracy with Major Token Savings**: On Llama3-8B (Table 5), PC achieves comparable or slightly higher accuracy than AC/ESC (e.g., 67.8% vs. 67.2% on GSM8K) while drastically reducing token usage—57.8% vs. -41.7% (AC) and -29.8% (ESC). Similar patterns appear on SVAMP and StrategyQA (-64.8% and -61.2%, respectively).
> - **Consistent Efficiency on Strong Models**: On DeepSeek-V3 (Table 6), PC matches AC/ESC accuracy across datasets while reducing tokens by 5–42%, showing that efficiency gains persist even with highly capable models.
> - **Strong Effectiveness on Smaller Models**: On Llama-3.2-1B (Table 7), PC uses roughly half the tokens of AC/ESC on GSM8K and StrategyQA with similar accuracy, demonstrating efficient use of limited computational resources.
>
> Mechanistically, the advantage of PC comes from preserving the diversity of reasoning paths while selectively committing to high-confidence prefixes. In contrast, AC and ESC accelerate inference primarily via early-exit strategies, which prune reasoning paths aggressively and can reduce diversity. By leveraging statistical confidence in prefixes rather than single-shot early exits, PC achieves a more efficient and reliable use of available samples.
>
>
>
> **References**:
>
> Xuezhi Wang, Jason Wei, Dale Schuurmans, Quoc V Le, Ed H Chi, Sharan Narang, Aakanksha Chowdhery, and Denny Zhou. Self-consistency improves chain of thought reasoning in language models. In *The Eleventh International Conference on Learning Representations*, 2023.
>
> Pranjal Aggarwal, Aman Madaan, Yiming Yang, et al. Let’s sample step by step: Adaptive-consistency for efficient reasoning and coding with llms. *arXiv preprint arXiv:2305.11860*, 2023.
>
> Yiwei Li, Peiwen Yuan, Shaoxiong Feng, Boyuan Pan, Xinglin Wang, Bin Sun, Heda Wang, and Kan Li. Escape sky-high cost: Early-stopping self-consistency for multi-step reasoning. In *The Twelfth International Conference on Learning Representations*, 2024.

---

### Official Review · Reviewer_uF6r · 2025-11-01

**Soundness:** 3
**Presentation:** 3
**Contribution:** 3
**Rating:** 4
**Confidence:** 3

**Summary:**

Paper proposes path-consistency with prefix enhancement for LLMs to solve the high computational cost of self-consistency. It extracts "prefixes" from early-generated paths using answer confidence to guide subsequent branch generation, reducing errors and token waste without extra computation, fine-tuning, or model changes. Evaluated on 10 datasets, it boosts inference speed and also outperforms baselines.

**Strengths:**

1.The paper demonstrates originality through a creative reformulation of self-consistency.  It does not merely prune paths but repurposes high-confidence segments to steer inference.
2. The work exhibits high methodological and empirical quality. Methodologically, PC is rigorously designed. Empirically, the evaluation is comprehensive.
3. The paper is well-structured and accessible. The "extract-and-sample" process is clearly illustrated through pseudocode and a running example, making the workflow easy to follow.
4. This work addresses the high computational cost of self-consistency in LLM deployment. By making inference drastically more efficient without accuracy loss, path-consistency has immediate practical significance for real-world applications requiring reliable reasoning.

**Weaknesses:**

1.The paper relies heavily on the beta confidence criterion but provides minimal analysis of how sensitive the results are to this specific choice.
2.The theoretical analysis assumes only one dominant incorrect answer, which oversimplifies real-world scenarios where multiple plausible but wrong answers compete.
3.While arithmetic/reasoning tasks show impressive gains, there's no evaluation on generation-intensive tasks (e.g., long-form QA, summarization) where prefix reuse might have different effects.

**Questions:**

1. Paper mentions extracting "shorter prefixes from optimal paths" when confidence exceeds a threshold, but it does not define what constitutes a "step" or how steps are segmented from raw model outputs. Is it rule-based or data-driven? Does step segmentation vary across tasks?
2. What is the computational overhead of PC’s prefix extraction and confidence calculation?
3. PC performs iterative prefix selection, not one-time selection. Does iterative selection increase the risk of amplifying early minority errors (e.g., a wrong prefix selected at Level-1 leading to more wrong paths at Level-2)?
4."randomly sample from the extracted prefixes as part of the prompt." What is the sampling strategy？

---

> ### Author Response · Authors · 2025-11-21
> **Response to Reviewer uF6r**
>
> Dear Reviewer uF6r,
>
> Thank you for the detailed and constructive feedback! We appreciate the opportunity to address your concerns and improve our work.
>
> > **Q1**: Step segmentation mechanism
>
> PC uses a simple, rule-based segmentation strategy: each step corresponds to one complete natural-language sentence produced by the model (identified via punctuation delimiters). This procedure is task-independent and does not rely on dataset-specific heuristics.
>
> > **Q2**: Computation overhead of PC
>
> We empirically evaluated the runtime cost of PC on GSM8K using Llama3-8B with 20-branch sampling per problem. The additional overhead of prefix extraction and Beta confidence evaluation was negligible compared to the model sampling time.
>
> | Component                  | Time (s) | Ratio to Generation Time |
> | -------------------------- | -------- | ------------------------ |
> | Model generation           | 2.2      | 100%                     |
> | Beta confidence evaluation | 0.000096 | 0.0044%                  |
> | Prefix extraction          | 0.00013  | 0.0059%                  |
>
> This demonstrates that PC adds only a tiny fraction of overhead relative to model inference, confirming that the efficiency gains from reduced token generation dominate any computation cost from prefix handling.
>
> > **Q3**: Risk of error amplification
>
> Iterative prefix selection **does not increase the risk of propagating early minority errors**. In fact, the probability that an incorrect prefix passes multiple Beta gates decreases exponentially with depth, since each gate independently requires a majority concentration above the threshold. We include a formal proof in the **Supplementary Theoretical Analysis** and will be added to the appendix to make this behavior explicit.
>
> We analyzed the per-branch behavior of PC and found that prefix correctness does not need to be perfect in early iterations, as the method naturally performs self-correction when early prefixes become unsupportive. Let's see this case:
>
> ```json
> {"input":"Two girls each got 1\/6 of the 24 liters of water. Then a boy got 6 liters of water. How many liters of water were left?","target":10}
> {"id": 80, "result": 1, "target": 10, "answer": 10.0,
> "answers": [8.0, 10.0, 14.0, 14.0, 14.0, 2.0, 14.0, 14.0, 10.0, 16.0, 10.0, 10.0, 10.0, 10.0, 12.0, 16.0, 14.0, 2.0, 10.0, 14.0],
> "tokens_gen": [70, 90, 63, 76, 84, 52, 57, 59, 54, 61, 15, 15, 15, 15, 29, 68, 65, 76, 94, 77], "
> level": [0, 0, 0, 0, 0, 1, 1, 1, 1, 1, 2, 2, 2, 2, 2, 0, 0, 0, 0, 0],
> ```
>
> In this GSM8K instance using the Llama-3.2-1B-Instruct model, the first several sampled branches do not converge to the correct answer, and the initially extracted prefix is itself incorrect. However, as the sampling branches expands, the distribution shifts, the confidence of the current prefix falls below the Beta-gated threshold, and PC **automatically discards** the erroneous prefix. This “prefix correction” step allows later branches to align toward the correct reasoning (final answer: 10 liters), despite early inconsistencies.
>
> > **Q4**: Sampling strategy
>
> We apply uniform random sampling over all extracted prefixes that satisfy the Beta gate. Because all candidates are already verified as high-confidence, uniform sampling avoids bias and keeps PC model-agnostic and efficient.
>
> > **W2**: Assumption on number of incorrect answer
>
> The assumption of a single dominant incorrect answer is not a simplification for convenience but the provable worst case for PC. Under our assumption in the paper, the model’s SC performance on the dataset yields an accuracy above 50%. Concentrating all wrong-probability mass onto one competitor maximizes the chance that a wrong prefix surpasses the Beta gate; distributing this mass across multiple answers only decreases this probability. We provide a formal proof in the **Supplementary Theoretical Analysis** to clarify this point.
>
> > **W3**: Generation-intensive tasks
>
> PC is primarily designed for short-form chain-of-thought scenarios where solution paths consist of semantically discrete reasoning steps. In such tasks, path-consistency effectively reduces unnecessary token generation and accelerates inference.
>
> To test PC on longer reasoning chains, we have added experiments on the AIME 2025 dataset using DeepSeek-V3, which produces long CoT sequences. The results show that while PC’s speedup effect remains present on long chains, it is less pronounced, achieving a 4% reduction in token consumption. Meanwhile, there is a substantial improvement in accuracy when applying PC, because of PC filtering out early unstable reasoning branches and enforcing prefix agreement, which helps long CoT sequences converge to more coherent and correct solutions. Thank you for the helpful suggestion.
>
> AIME2025:
>
> | Method | Accuracy (%) | Decrease (%) |
> | ------ | ------------ | ------------ |
> | SC     | 60.0         | -            |
> | PC     | 76.7         | -4.1%        |

---

### Official Review · Reviewer_YQkP · 2025-11-02

**Soundness:** 3
**Presentation:** 2
**Contribution:** 2
**Rating:** 6
**Confidence:** 3

**Summary:**

The authors propose Path-Consistency (PC), an inference-time method that accelerates Self-Consistency (SC) decoding for LLMs without additional training or model changes.
PC repeatedly (i) generates a small window of reasoning paths, (ii) estimates answer confidence, (iii) extracts the common prefix of high-confidence paths, and (iv) re-uses that prefix to seed the next window.
Iterating this “extract-and-sample” loop shortens subsequent generations, cutting token cost and wall-clock latency while preserving (or slightly improving) accuracy on ten reasoning benchmarks.
Extensive experiments with Llama-3-8B show up to 48 % speed-up and −58 % token consumption vs. vanilla SC, with comparable or better accuracy on math, commonsense, and symbolic tasks. The method is model-agnostic and orthogonal to other SC optimizations.

**Strengths:**

1. Significant efficiency gains: Demonstrates 20–48 % latency reduction and 20–60 % fewer tokens across diverse datasets, verified on a consumer GPU.
2. No training or architecture change: Pure inference algorithm—easy to plug into any decoder-only LLM and complementary to distillation, early-exit, or speculative schemes.
3. Careful ablation & analysis: Authors vary confidence thresholds, prefix levels, model sizes, and datasets; provide theoretical justification that PC does not amplify “truth-in-the-hands-of-a-few” problem when base SC already works.

**Weaknesses:**

1. Limited novelty: Core idea—early-commitment to promising partial solutions—closely resembles existing beam/pruning and adaptive-consistency techniques; incremental contribution.
2. Heavily engineered hyper-parameters: Window size, #branches, confidence metric, prefix level schedule, and threshold all require per-task tuning; no automatic or adaptive schedule offered.
3. Evaluation restricted to 20-path SC: All speed/accuracy numbers compare against a 20-sample baseline; unclear how PC behaves with larger budgets or stronger teacher models where SC accuracy saturates.

**Questions:**

1. How does PC compare when SC uses 40, 100, or adaptive samples—does the relative speed-up shrink or the error rate rise?
2. Fig. 2 shows 25–50 % tokens wasted on wrong branches; does aggressively lengthening the prefix (level-4/5) ever mis-guide later samples and hurt accuracy?
3. The proof assumes Pvote ≥ 0.5; what bound can you give for harder datasets where SC itself is below 50 %, and how should practitioners set Cthreshold in that regime?

---

> ### Author Response · Authors · 2025-11-21
> **Response to Reviewer YQkP**
>
> **Dear Reviewer YQkP,**
>
> We thank you for your thoughtful and constructive feedback. We sincerely appreciate the opportunity to clarify our contributions and address the raised concerns.
>
> > **Q1&W3**: Evaluation restricted to 20-path
>
> We chose the 20-path SC baseline because prior work (e.g., Fig. 2 in Wang et al., 2023) shows that accuracy gains from increasing the number of SC samples rapidly saturate: moving from 20 to 40 or even 80 samples yields only marginal improvements. Thus 20-path is already close to the plateau of SC performance.
>
> Importantly, PC is designed as an **efficiency** method. Using 20-path as the baseline is actually **more demanding** for our approach: the fewer branches SC uses, the harder it is for PC to match its accuracy with reduced sampling. In contrast, evaluating PC against 40+ samples would make the task easier, not harder.
>
> However, we have added experiments using 40-path SC and PC on GSM8K with Llama-8B:
> | Method | Accuracy (%) | Tokens (# per problem) |
> | ------ | ------------ | ---------------------- |
> | SC-40  | 68.9         | 3861                   |
> | PC-40  | 68.5         | 1988 (-48.5%)          |
>
> > **Q2**: Effect of higher prefix levels
>
> As shown in **Table 4** (GSM8K):
> | Method     | Level | Accuracy (%)       | Speedup (%)   | Token Decrease (%) |
> | ---------- | ----- | ------------------ | ------------- | ------------------ |
> | PC (C=0.7) | 3 → 4 | 67.1 → 66.3 (–0.8) | +18.0 → +20.5 | –16.1 → –20.0      |
> | PC (C=0.8) | 3 → 4 | 67.8 → 67.8 (+0.0) | +17.4 → +18.0 | –16.2 → –17.8      |
>
> increasing the prefix level from 3 to 4 results in only a negligible accuracy change (–0.8% for C=0.7 and exactly 0% for C=0.8), while both speedup and token reduction consistently improve. This indicates that more aggressive prefix extraction does not amplify errors; instead, it further reduces unnecessary computation without harming correctness. Overall, deeper prefix gating remains stable and continues to provide efficiency gains without degrading accuracy.
>
>
> > **Q3**: Assumption on model performance
>
> PC is a model-agnostic efficiency method. It reduces latency and token usage, but does not change improve raw model quality. If the base model performs poorly (SC < 50%), PC will still provide efficiency gains, but cannot correct systematic model errors. We analyze two cases:
>
> | Regime                     | Base Self-Consistency | Effect of PC                                                 | Guidance for C_threshold                                     |
> | -------------------------- | --------------------- | ------------------------------------------------------------ | ------------------------------------------------------------ |
> | **High-confidence regime** | P_vote ≥ 0.5          | PC accelerates inference without degrading accuracy; aggressiveness may trade small accuracy for large speedups | Smaller C_threshold to maximize efficiency                   |
> | **Low-confidence regime**  | P_vote < 0.5          | Model is unreliable regardless of sampling; PC still saves cost | Larger C_threshold (e.g., 0.8) recommended to avoid over-committing to noisy early prefixes |
>
> Practically, path-consistency is an **efficiency** method, not a technique for improving a fundamentally low-performing model. If a dataset is so difficult that SC < 50%, the priority is to improve the model’s reasoning ability rather than to apply PC, which focuses purely on efficiency gains once a reasonable baseline is established.
>
>
> > **W2**: Heavily engineered hyper-parameters
>
> - **Window size**: The window size is not a free parameter; it is structurally tied to the number of branches and the maximum prefix level:
>
> $$
> Window\\_size = number\\_of\\_branches // maximum\\_prefix\\_level
> $$
>
> - **Number of branches**: This parameter is inherent to the SC baseline, not introduced by PC. The SC paper itself explores this parameter (e.g., Fig. 2 in Wang et al., 2023), showing saturation around 20 branches. We follow that established operating point.
>
> - **Confidence metric**: Although several confidence metrics are possible, we used the Beta metric because it is:
>   - analytically tractable for our theoretical guarantees,
>   - computationally lightweight,
>   - and empirically stable.
>
> - **Prefix level schedule&threshold**: These two parameters control the efficiency–accuracy trade-off:
>   - Smaller thresholds or deeper prefix levels -> more aggressive extraction (higher speedup).
>   - Larger thresholds or shallower levels -> more conservative extraction (higher accuracy).
>     Notably, when the threshold is below 0.5, PC reduces to majority voting, giving a clean conceptual baseline.
>
> **References**:
>
> Wang, X., Wei, J., Schuurmans, D., Le, Q., Chi, E., Narang, S., Chowdhery, A., & Zhou, D. (2023). Self-consistency improves chain of thought reasoning in language models. arXiv preprint arXiv:2203.11171.

---

### Author Response · Authors · 2025-11-21
**Supplementary Theoretical Analysis: Soundness of Beta-Gated Prefix Selection**

## **Theorem 1** (Single-Level Soundness of Beta-Gated Prefix Selection) Consider a fixed level of prefix extraction with the following setup:
$\mathcal A$ denotes a finite answer set
The algorithm performs promotion checks at a sequence of increasing sample counts:
$$0 < n^{(1)} < n^{(2)} < \dots < n^{(K)} = n_{\max}$$
At each check $k$, we draw the first $n^{(k)}$ i.i.d. samples:
$$X_1, \dots, X_{n^{(k)}}\in\mathcal A$$
For each $a\in\mathcal A$, let
$$\pi(a) := \Pr(X_i = a)$$
denote the model's sampling probability under the prefix.
The correct answer is $a^\star$ with:
$$q := \pi(a^\star), R := 1-q = \sum_{a \neq a^\star} \pi(a)$$
For each check $k$, define the observed counts:
$$N_a^{(k)} := \sum_{i=1}^{n^{(k)}} \mathbf{1}\\{X_i = a\\}, \hat a^{(k)} := \arg\max_{a\in \mathcal A} N_a^{(k)}, s^{(k)} := N_{\hat a^{(k)}}^{(k)}$$
Assume a Beta-gated promotion threshold $\tau \ge 0.5$, where promotion occurs at check $k$ if:
$$T(s^{(k)},n^{(k)}) := \Pr(\theta > 1/2 \mid \text{Beta}(s^{(k)}+1,\,n^{(k)}-s^{(k)}+1)) > \tau$$

Then, if $q \ge 1/2 + \Delta$ for some $\Delta>0$, the probability that the correct prefix fails to be promoted at check $k$ is bounded by:
$$\Pr(A_k) \le \exp\big(-2 n^{(k)} \Delta^2\big)$$
where $A_k := \{\text{not promoted at check } k\}$.

*proof*. Promotion at check $k$ requires that $\hat a^{(k)} = a^\star$ and $T(s^{(k)},n^{(k)})>\tau$. Thus failure at check $k$ implies:
$$A_k \subseteq \{ N_{a^\star}^{(k)} < k_{\min}(n^{(k)},\tau) \}$$
where:
$$k_{\min}(n,\tau) := \min\{r \in \{0,\dots,n\}: T(r,n)>\tau\}$$

Since $\tau\ge 0.5$, symmetry of the Beta posterior implies:

$$k_{\min}(n^{(k)},\tau) > n^{(k)}/2$$

Hence:

$$A_k \subseteq \{N_{a^\star}^{(k)} \le n^{(k)}/2\}$$

But $N_{a^\star}^{(k)}$ is a sum of $n^{(k)}$ independent Bernoulli($q$) variables with mean at least $n^{(k)}(1/2+\Delta)$. By Hoeffding's inequality:

$$\begin{aligned}
\Pr\Big(N_{a^\star}^{(k)} \le n^{(k)}/2\Big) &= \Pr\Big(N_{a^\star}^{(k)} \le \mathbb{E}[N_{a^\star}^{(k)}] - n^{(k)}\Delta \Big) \\
&\le \exp\big(-2 n^{(k)} \Delta^2\big)
\end{aligned}$$

This completes the proof.

## **Corollary 1** (Multi-Level Promotion Soundness)
Consider $m$ consecutive prefix-extraction levels indexed by $\ell_1,\dots,\ell_m$. At each level $\ell_j$, promotion checks occur at sample sizes $n_{\ell_j}^{(1)} < \cdots < n_{\ell_j}^{(K_j)}$, with correct-answer probabilities $q_{\ell_j} \ge 1/2 + \Delta_{\ell_j}$ and Beta thresholds $\tau_{\ell_j} \ge 0.5$. Let $A_{\ell_j}^{(k)}$ denote the failure-to-promote event at the $k$-th check of level $\ell_j$. Then, under conditional independence across levels,
\\[\Pr(\text{failure along full promotion path})\le\prod_{j=1}^m\Bigg(\sum_{k=1}^{K_j} \exp\big(-2 n_{\ell_j}^{(k)} \Delta_{\ell_j}^2\big)\Bigg).\\]

*proof.* For each level $\ell_j$, we assume the true success probability remains $q_{\ell_j} \ge 1/2 + \Delta_{\ell_j}$. This assumption is valid due to the one-time selection guarantee proved in Section~4.2, which ensures that the prefix extraction does not hurt accuracy. Thus, prefix quality is not degraded by previous selection decisions; it preserves a positive advantage $\Delta_{\ell_j}$ at every level.
At each promotion check $k$ of level $\ell_j$, requiring posterior probability

\\[
\Pr(\theta_{\ell_j} > 1/2) > \tau_{\ell_j} \ge 0.5,
\\]

Hoeffding’s bound implies that the probability that a truly good prefix fails to promote at this check satisfies

\\[
\Pr(A_{\ell_j}^{(k)}) \le \exp(-2n_{\ell_j}^{(k)} \Delta_{\ell_j}^2).
\\]

Since promotion only needs to succeed once within a level, the probability of failure across all checks of level $\ell_j$ is bounded by

\\[
\Pr\Big(\bigcap_{k} A_{\ell_j}^{(k)}\Big)
\le
\sum_{k=1}^{K_j} \exp(-2n_{\ell_j}^{(k)} \Delta_{\ell_j}^2).
\\]

Finally, assuming conditional independence across levels,

\\[
\Pr(\text{failure along full promotion path})=
\Pr\Big(\bigcap_{j=1}^m \bigcap_{k=1}^{K_j} A_{\ell_j}^{(k)}\Big)
\le
\prod_{j=1}^m
\Bigg(
\sum_{k=1}^{K_j} \exp(-2 n_{\ell_j}^{(k)} \Delta_{\ell_j}^2)
\Bigg).
\\]

Thus the overall probability of failing to amplify a prefix with true advantage $\Delta_{\ell_j}>0$ decays exponentially with the total effective sample budget across levels.

## **Corollary 2** (Worst-Case Wrong Answer Distribution)
For fixed total wrong-mass $R = 1-q$, the probability that a wrong answer is promoted is maximized when all wrong probability is concentrated on a single alternative. In that case, the probability of wrong promotion is
\\[
\Pr(N_{a_\text{bad}} \ge k_{\min}(n,\tau)), \qquad a_\text{bad} \in \mathcal A \setminus \{a^\star\}.
\\]
*proof.* Let $T := \sum_{a \neq a^\star} N_a$ denote the total count of wrong answers. Then
\\[
\{\exists a\neq a^\star: N_a \ge k_{\min}\} \subseteq \{T \ge k_{\min}\}.
\\]
When the entire wrong-mass $R$ is assigned to a single $a_\text{bad}$, $T = N_{a_\text{bad}}$, and the inclusion becomes equality. Hence the probability of wrong promotion is maximized in this configuration.

---

### Meta-Review · Area_Chair_hmL7 · 2025-12-29

**Summary:**

This paper proposes Path-Consistency (PC), an inference-time variant of self-consistency that reuses high-confidence prefixes to reduce token consumption. While the idea is reasonable and the paper is clearly written, the current evidence does not support acceptance at ICLR. Primary concerns:
- Limited and inconsistent empirical support across models.
 The main results are heavily centered on LLaMA-3-8B (base), where PC shows moderate efficiency gains. However, on DeepSeek-V3 (Table 6) and LLaMA-3.2-1B-Instruct (Table 7), PC does not show a clear accuracy advantage over AC or ESC, and often performs similarly. Moreover, self-consistency (SC) results are missing in Tables 6 and 7, making it difficult to assess the true trade-offs. The choice of using LLaMA-3-8B (base) rather than the much stronger LLaMA-3-8B-Instruct—while using an instruct model for the small-scale setting—is insufficiently justified and weakens the empirical narrative.
- Insufficient comparison with modern reasoning baselines.
 The experimental comparisons are largely restricted to SC, AC, and ESC. As pointed out by Reviewer XE47 (W1), self-consistency is no longer a strong or complete baseline, and more recent inference-time reasoning methods (e.g., Tree-of-Thoughts, self-verification, Graph-of-Thoughts, Self-Refine, Reflexion-style methods) are absent. The rebuttal does not convincingly resolve this gap, leaving the paper’s relevance to current reasoning literature unclear.
- Methodologically, PC relies on several hand-designed and task-sensitive components (window size, prefix level schedule, confidence threshold), yet provides no principled or adaptive mechanism for setting them. In addition, prefix extraction based on sentence-level heuristics risks mis-segmenting or over-committing to spurious early reasoning steps, particularly for long or heterogeneous chains of thought. These issues limit robustness and generality beyond the evaluated settings.

 Although the paper presents a reasonable efficiency-oriented variant of self-consistency, its empirical advantages are narrow, its baseline coverage is outdated, and its methodological assumptions and hyperparameter sensitivity are not sufficiently addressed. Given these unresolved concerns, especially regarding evaluation on stronger models and against more modern reasoning methods, the paper does not meet the bar of ICLR 2026. I therefore tend to reject this paper.

**Reviewer Concerns:**

The rebuttal clarifies efficiency measurement, reports additional latency numbers, adds limited extra experiments, and explains implementation details such as step segmentation and negligible overhead.

However, the core concerns remain unresolved: results are still centered on a single base model with no clear advantage on stronger or instruction-tuned models, some key self-consistency results are missing, comparisons with modern reasoning methods are absent, and the method continues to rely on hand-tuned hyperparameters and heuristic prefix segmentation, limiting robustness and generality.

**Reviewer Scores:**

4,4,6,6

---

### Decision · Program_Chairs · 2026-01-26

Reject